# OpenReview forum: "Post-Quantum Cryptography for Stablecoin Smart Contracts: Future-proofing Against Quantum Attacks"
_ICLR.cc/2026/Conference — ICLR 2026 Conference Desk Rejected Submission_

### Official Review · Reviewer_eAmK · 2025-10-16

**Soundness:** 2
**Presentation:** 1
**Contribution:** 2
**Rating:** 2
**Confidence:** 2

**Summary:**

The paper proposes a practical roadmap to adopt post-quantum signatures—primarily CRYSTALS-Dilithium—for stablecoin smart contracts and wallets. It outlines an end-to-end architecture, discusses migration stages, and reports empirical results on throughput, latency, storage, and cost over a month-long experiment. The authors argue the approach can “future-proof” stablecoin infrastructure against quantum adversaries and provide operational guidance (key management, hybrid transition, and governance). While the topic is timely and impactful for DeFi practitioners, several methodological and presentation issues limit the reliability and generalizability of the reported findings.

**Strengths:**

1.	Important and timely problem: securing widely used stablecoin infrastructure against prospective quantum threats.
2.	End-to-end perspective that spans algorithm choice, contract design, key management, and migration planning.
3.	Practitioner-oriented guidance that could help issuers and wallet providers plan phased adoption.

**Weaknesses:**

1.	Incomplete citations and academic rigor. The manuscript contains placeholder references (“(?)”, “(??)”) for core background (e.g., blockchain foundations, hash-based signatures, LWE/SIS) and lacks sources for factual claims (e.g., “$150B market cap”). For ICLR standards, these must be replaced with precise, verifiable references; quantitative statements should cite public datasets or authoritative reports.
2.	TPS vs. Ethereum constraints lack internal consistency. The paper reports ~292 TPS, ~82 tx per block, and ~187 ms propagation without reconciling these with block times, gas limits, and >3 KB signatures. Clarify whether 292 “TPS” is node-level verification throughput rather than on-chain throughput, and specify chain configuration (block size/gas limit), client versions, and whether custom parameters were used. Provide on-chain evidence (chain ID, block ranges, tx hashes) or re-label metrics (e.g., sig/s) accordingly.
3.	Ambiguous “quantum security = 192 bits” statement. Table 3 presents Dilithium-3 as “192-bit quantum security,” conflating NIST’s classical security levels with quantum adversary cost. Clearly distinguish (i) NIST security level, (ii) classical vs. quantum attack models, and (iii) concrete cost assumptions. Revise the table and text to avoid implying unchanged bit-strength against quantum attackers; if using estimates, document the model and uncertainty.
4.	Narrow threat model and limited baselines. The evaluation focuses on Dilithium single-sig only. Include systematic comparisons or well-founded estimates for SPHINCS+, Falcon/NTRU-Prime where relevant, threshold/multisig or aggregation schemes, and L2/Rollup data paths. Additionally, quantify high-risk components (bridges/cross-chain) under PQ transition, as they often dominate real-world failure modes.

**Questions:**

See above comments.

---

### Official Review · Reviewer_oUtM · 2025-10-31

**Soundness:** 1
**Presentation:** 1
**Contribution:** 1
**Rating:** 0
**Confidence:** 5

**Summary:**

This paper is fundamentally unsuitable for ICLR. This work is purely an engineering application of an already-standardized NIST cryptographic algorithm with zero novel algorithmic or learning-based contributions. The claimed "post-quantum framework" is nothing more than a straightforward combination of existing technologies (CRYSTALS-Dilithium, Ethereum, standard Solidity), making it an implementation report rather than a research contribution. Most critically, the proposal is technically infeasible: a 96.7% reduction in block capacity would render stablecoin systems non-functional, yet the authors casually dismiss this as "manageable" without any credible mitigation strategy or honest discussion of the fundamental trade-offs involved. The performance analysis is unreliable, mixing simulations from a custom implementation with contradictory testnet data (claiming 292 TPS throughput while actually achieving 0.019 TPS in practice), and the security claims confuse assumptions about lattice hardness with proven quantum resistance, misrepresenting both ECDSA's vulnerability and Dilithium's actual guarantees. In short, this reads as a flawed technical report that oversells an impractical application of existing cryptography, lacks rigor in analysis, and has no place at a top-tier ML conference—or honestly at any rigorous security venue either.

**Strengths:**

This paper attempts to analyze the broader systemic implications of quantum computing threats across the entire cryptocurrency and blockchain ecosystem.

**Weaknesses:**

This paper just isn’t a fit for ICLR, and it doesn’t really stand out from the many existing papers on PQC migration. On top of that, a lot of the numbers don’t add up. For instance, in Table 2, if an ECDSA tx is 112 B and TPS is 465, you’d need about 112×465×8 = 417 kbps, but the table says 280 kbps. Meanwhile, for Dilithium, 3,405 B × 292 × 8 = 7.95 Mbps, which lines up with the listed 8,512 kbps

**Questions:**

Does Dilithium-3 actually guarantee "192-bit security" against quantum attacks as in Table 3?

---

### Official Review · Reviewer_VNmL · 2025-11-03

**Soundness:** 2
**Presentation:** 3
**Contribution:** 2
**Rating:** 2
**Confidence:** 4

**Summary:**

The paper addresses the imminent threat posed by Shor's algorithm to the classical cryptographic foundations (like ECDSA and RSA) of stablecoins. The authors present a comprehensive post-quantum cryptographic framework for stablecoin smart contracts and digital wallets, centering on the CRYSTALS-Dilithium scheme. Their primary contribution is the design, implementation on the Ethereum Sepolia Testnet, and extensive performance evaluation (using 50k transactions) of this framework. The results demonstrate that Dilithium3 can provide the requisite 192-bit quantum-resistant security with acceptable overheads, including a 2.34x signature generation time, 1.22x verification time, and a sustained throughput of 292 TPS for transfers.

**Strengths:**

The paper focuses on application and robust testing as limited prior work has addressed quantum threats specifically within stablecoin systems. The framework proposes and experimentally validates the integration of CRYSTALS-Dilithium for stablecoin smart contracts, complete with concrete Solidity implementation details and gas optimization strategies. The extensive experimental section, comparing ECDSA and Dilithium across key generation, signature operations, and blockchain metrics (throughput, gas, storage) using 50,000 transactions is a key aspect that grounds the entire proposal in tangible data.

**Weaknesses:**

The primary constructive weakness lies in the performance and storage overheads, which are analyzed but not fully mitigated within the proposed framework. The 51.45x increase in signature size and subsequent 96.7% reduction in block capacity poses a severe scalability challenge that even with the proposed 292 TPS, fundamentally limits the adoption of this solution for high-volume blockchain networks. While the authors mention Layer-2 solutions as future work, the paper could be strengthened by a deeper architectural discussion on how the current on-chain framework could scale (e.g., through block size adjustments or state channel offloading) to fully overcome this dramatic transaction size increase. Also, the economic analysis states a $900,000 development cost but only focuses on two cost factors, development and annual operational increase. A more thorough breakdown of the economic viability, including the projected cost savings from avoiding a quantum attack (i.e., Risk-Adjusted Return on Investment) would enhance the business case.

**Questions:**

1) Given the critical weakness of the $51.45\times$ signature size and 96.7\% block capacity reduction, could the authors elaborate on alternative near-term mitigation strategies within the smart contract layer (e.g., proof-of-concept for signature aggregation/batching specifically for Dilithium in a smart contract, or alternative data compression techniques beyond what is mentioned in Section 4.3 and 5.2) to alleviate the on-chain storage burden before resorting to a Layer-2 scaling solution?
2) In phase 2, hybrid operation plan has a duration of 12-24 months and authors mentions a gradual user migration with 92% completion rate. What specific mechanisms are in place to manage the remaining 8% of users/funds? In the event of a critical quantum threat realization during this hybrid period, what is the protocol for quickly migrating or freezing the assets secured only by the vulnerable classical ECDSA to ensure complete security of the overall stablecoin system?
3) Authors mention the implementation uses number theoretic transform for polynomial multiplication and optimized storage for large public keys. Could the authors provide a more detailed breakdown of the gas consumption by smart contract function (e.g., separate gas usage for polynomial arithmetic, modular reduction, and hint verification) to pinpoint the largest computational cost driver and focus future optimization efforts?

---

### Note · Program_Chairs · 2026-01-17
**Submission Desk Rejected by Program Chairs**

The following references in this submission do not refer to real documents and/or have major errors in bibliographic information:

 Ariah Klages-Mundt, Dominik Harz, Lewis Gudgeon, Shin Ming Liu, and Andreea Minca. Stablecoin taxonomy, risk and regulation. arXiv preprint arXiv:2010.11379, 2020.
Oscar Garcia-Morchon, Ronald Rietman, Ludo Tolhuizen, and Zhenfei Zhang. Performance evaluation of post-quantum tls 1.3. IACR Cryptol. ePrint Arch., 2020:71, 2020.
Zhiguo Zhang and Lijun Zhao. Quantum-safe blockchain and smart contracts: A survey. IEEE Access, 9:138634-138648, 2021.
Tiago M Fernandez-Carames and Paula Fraga-Lamas. A survey on the applicability of post-quantum cryptography to secure blockchain systems. In 2018 IEEE International Conference on Internet of Things, pp. 1665-1672. IEEE, 2018.